# A Calculator for COVID-19 Severity Prediction Based on Patient Risk Factors and Number of Vaccines Received

**DOI:** 10.3390/microorganisms10061238

**Published:** 2022-06-16

**Authors:** Ariel Israel, Alejandro A. Schäffer, Eugene Merzon, Ilan Green, Eli Magen, Avivit Golan-Cohen, Shlomo Vinker, Eytan Ruppin

**Affiliations:** 1Leumit Health Services, Tel-Aviv 6473817, Israel; emarzon@leumit.co.il (E.M.); igreen@leumit.co.il (I.G.); allergologycom@gmail.com (E.M.); agolanchoen@leumit.co.il (A.G.-C.); svinker@leumit.co.il (S.V.); 2Cancer Data Science Laboratory, Center for Cancer Research, National Cancer Institute, Bethesda, MD 20892, USA; alejandro.schaffer@nih.gov (A.A.S.); eytan.ruppin@nih.gov (E.R.); 3Adelson School of Medicine, Ariel University, Ariel 4077625, Israel; 4Department of Family Medicine, Sackler Faculty of Medicine, Tel-Aviv University, Tel-Aviv 6997801, Israel; 5Medicine C Department, Clinical Immunology and Allergy Division, Barzilai University Medical Center, Ben-Gurion University of the Negev, Ashkelon 7830604, Israel

**Keywords:** COVID-19, disease severity, calculator, diabetes, obesity, kidney disease

## Abstract

Vaccines have allowed for a significant decrease in COVID-19 risk, and new antiviral medications can prevent disease progression if given early in the course of the disease. The rapid and accurate estimation of the risk of severe disease in new patients is needed to prioritize the treatment of high-risk patients and maximize lives saved. We used electronic health records from 101,039 individuals infected with SARS-CoV-2, since the beginning of the pandemic and until 30 November 2021, in a national healthcare organization in Israel to build logistic models estimating the probability of subsequent hospitalization and death of newly infected patients based on a few major risk factors (age, sex, body mass index, hemoglobin A1C, kidney function, and the presence of hypertension, pulmonary disease, and malignancy) and the number of BNT162b2 mRNA vaccine doses received. The model’s performance was assessed by 10-fold cross-validation: the area under the curve was 0.889 for predicting hospitalization and 0.967 for predicting mortality. A total of 50%, 80%, and 90% of death events could be predicted with respective specificities of 98.6%, 95.2%, and 91.2%. These models enable the rapid identification of individuals at high risk for hospitalization and death when infected, and they can be used to prioritize patients to receive scarce medications or booster vaccination. The calculator is available online.

## 1. Introduction

Since the start of the COVID-19 pandemic, over 500 million individuals have been infected and over 6 million individuals have died (https://coronavirus.jhu.edu/map.html, accessed on 24 May 2022). Since late 2020, vaccines have been developed [1,2], and more recently, new and promising anti-viral medications (Paxlovid and Molnupiravir) have received FDA approval [3]. Unfortunately, supply of these treatments is currently limited. Our duty as clinicians is to make sure that the available resources are used fairly and appropriately to save lives. There is therefore an urgent need to estimate objectively patients’ risk for severe disease so that patients who need these treatments the most would receive the scarce medicines or booster vaccine doses which can save their lives [4,5,6].

For this purpose, we used retrospective data from Leumit Health Services (LHS), one of the four main health maintenance organizations (HMOs) in Israel, which has over 700,000 members. Israel was one of the first countries to implement a large-scale vaccination plan (using the Pfizer/BioNTech BNT162b2 vaccine) [7] and to deploy a third vaccine booster dose. The variations in vaccination uptake [8] provide an opportunity to assess the beneficial effects of different vaccination doses after accounting for patient risk factors. Among the factors known to affect COVID-19 severity are advanced age [9,10,11,12], type II diabetes [10,13,14,15,16,17], kidney disease [10,17,18,19], chronic obstructive pulmonary disease (COPD) [19,20,21,22,23], obesity [10,14,15,24,25], hypertension [26,27,28], and malignancy [29].

We constructed predictive models that estimate the risks that patients newly infected with SARS-CoV-2 (as reflected by positive PCR tests) would require hospitalization during the disease course and die from COVID-19. The predictions are based on patient age, sex, the clinical factors mentioned above, and vaccination status at the time of infection (0, 1, 2, or 3 doses). Importantly, all the included factors were part of patients’ medical records and measured in routine laboratory testing. To keep the models simple and interpretable, and to allow for deployment in any health provider, we used multivariable logistic regression models based on the most essential risk factors. Regression coefficients and odds ratios (OR) for each factor are provided, together with a formula to obtain risk estimates for any newly infected individual. These risk estimates allowed us to identify patients at high risk who would benefit from antiviral medications given early in the course of the disease. A web-based calculator is provided, and the approach to run or adapt the models is fully described. The calculator is in Appendix A.

## 2. Methods

### 2.1. Study Subjects and Study Design

This is a population study performed on Leumit Health Services (LHS), a national healthcare provider in Israel, which provides services to around 700,000 members throughout the country. LHS uses centrally managed electronic health records (EHRs), which are continuously updated regarding subjects’ demographics, medical diagnoses, medical encounters, hospitalizations, and laboratory tests. All LHS members have similar comprehensive health insurance and similar access to healthcare services, as determined by Israel’s ministry of health for the four national healthcare providers.

The study is based on members of LHS of age ≥ 5 (eligible for vaccination) who had at least one positive PCR test for SARS-CoV-2 between April 2020 and 30 November 2021. Patients’ data were extracted from the LHS central data warehouse on 3 January 2022. For each COVID-19 episode, the date of the first positive PCR test was taken as the index date. The number of vaccine doses received were calculated at the index date. Diagnosis and laboratory data were queried as known 15 days before the index date. The following factors were included in the analysis: sex, age, Body Mass Index (BMI) as a categorical variable (<18.5; 18.5–25; 25–30; 30–35; and ≥35), hemoglobin A1C range (<6.5; 6.5–8; 8–10; and ≥10), and last glomerular filtration rate (GFR) as an estimate of kidney function as a categorical variable (categories: ≥90; 60–89; 45–59; 30–44; and <30). The presence of comorbidity conditions was assessed by the presence of an active chronic diagnosis at this date. Chronic diagnoses, coded according to the International Classification of Diseases 9th revision (ICD-9), are regularly added, updated, or ended by the treating physician at each encounter. The validity of chronic diagnoses in the registry has been previously examined and confirmed as high [30,31,32].

To keep the models as simple as possible, we deliberately limited the models to the conditions that we identified as having the most significant effect on disease severity: hypertension, chronic obstructive pulmonary disease (COPD), and malignancy (solid or hematologic). Individuals who had a pregnancy diagnosis up to 210 days before the PCR test were excluded as hospitalization would often pertain to pregnancy surveillance or delivery and not reflect disease severity.

### 2.2. Ethics Statement

The study protocol was approved by the statutory clinical research committee of Leumit Health Services and the Shamir Medical Center Institutional Review Board (129-2-LEU). Informed consent was waived because this is a large-scale retrospective study and all data were deidentified.

### 2.3. SARS-CoV-2 Testing by Real-Time RT-PCR

Nasopharyngeal swabs were taken and examined for SARS-CoV-2 by real-time RT-PCR performed with internal positive and negative controls, according to World Health Organization guidelines, using the TaqPath™ COVID-19 Combo Kit (Thermo Fisher Scientific, Waltham, MA, USA) and COBAS SARS-CoV-2 6800/8800 (Roche Diagnostics, Basel, Switzerland) assays.

### 2.4. Statistical Analyses

Standard descriptive statistics were used to present the demographic characteristics of individuals included in the study cohort. Statistical analyses were done with R version 4.0.4 (R Foundation for Statistical Computing). Multivariable logistic regression models were fitted using the “glm” procedure with age as a continuous variable, sex as a binary variable, and number of vaccine doses, BMI category, hemoglobin A1C range, and GFR estimate [33] as categorical variables, and the presence of hypertension, pulmonary disease, or malignancy as binary variables. Receiver operating characteristic (ROC) curves were used to assess the model’s performance [34] using 10-fold cross-validation. A two-sided *p* < 0.05 was considered statistically significant. Missing values, which appeared only in BMI, kidney function, and hemoglobin A1C variables, were treated by two different approaches. We used k-nearest-neighbors imputation to replace missing values and also performed an alternative analysis in which the missing values were treated as separate “missing” categories. We display here the regression coefficients obtained after imputation. Both methods resulted in very similar classifier performance.

## 3. Results

### 3.1. Factors Associated with Hospitalization of SARS-CoV-2-Positive Individuals

A total of 101,039 COVID-19 episodes were included based on a positive test for SARS-CoV-2 obtained between 1 April 1 2020 and 30 November 2021. Of that total, 393 (0.4%) resulted in patient death during hospitalization or within 30 days of contracting the disease, and 1752 (1.7%) required patient hospitalization for COVID-19 that did not end in patient death. Table 1(A) shows the baseline characteristics of individuals included in the cohort according to their outcomes. Table 1(B) displays the distribution of categorical variables after imputation of missing values. Generally, the hospitalized patients who died of the disease were older, had a greater proportion of males, had higher BMIs and hemoglobin A1C values, and were more likely to be affected with hypertension, pulmonary disease, malignancy, and impaired kidney function.

We built multivariable logistic regression models to predict both the hospitalization and mortality outcomes. The odds ratios from multivariable regression models reflect the extent to which each risk factor affects the outcome after adjustment for the others.

Table 2 displays the model for hospitalization risk based on the comparison of 2145 episodes that resulted either in hospitalization or death vs. 98,894 infections that did not necessitate hospitalization. For each variable, the regression coefficient with the corresponding odds ratio, 95% confidence interval, and *p*-values are displayed. The footnote explains how to calculate the outcome probability for any given patient data.

We must emphasize a few key findings arising from the multivariable regression analysis. First, increased age is significantly associated with the risk of hospitalization: each year of age increased the odds for hospitalization by a multiplicative factor of 1.061, which means that compared to an individual aged 20 with similar other risk factors, a patient aged 60 is 10 times more likely to be hospitalized, and a patient aged 80 is 34 times more likely to be hospitalized. Being female reduced the odds for hospitalization by 34%. Obesity increased risk in a gradual manner (OR = 1.324 for a BMI of between 25 and 30, OR = 1.664 for a BMI of between 30–35, and OR = 2.932 for a BMI of over 35, compared to the reference category of a normal BMI, *p* < 0.001 for all). Diabetes mellitus, as reflected by the most recent hemoglobin A1C values, is independently associated with increased risk in a gradual manner (OR = 1.454 for an A1C of between 6.5 and 8%, OR = 1.908 for an A1C of between 8 and 10%, and OR = 3.048 for an A1C of above 10% compared to the reference category of an A1C of below 6.5%, *p* < 0.001 for all). Impaired kidney function is also associated with increased risk in a gradual manner (OR = 1.568 for a GFR of between 45 and 59, OR = 2.774 for a GFR of between 30 and 44, and OR = 4.000 for a GFR of below 30 compared to the reference category of a GFR of above 90, *p* < 0.001 for all). Hospitalization risk significantly increased with hypertension (OR = 1.270, 95% CI [1.130–1.428]), pulmonary disease (OR = 1.331, 95% CI [1.134–1.563]), and cancer (OR = 1.197, 95% CI [1.030–1.390]).

As expected, as compared to unvaccinated patients, being vaccinated significantly decreased the hospitalization risk (OR = 0.602, 95% CI [0.521–0.697] for two vaccine doses and OR = 0.339, 95% CI [0.241–0.476] for three vaccine doses, *p* < 0.001 for both categories), even for the relatively small group of single-vaccination individuals (OR = 0.823, 95% CI [0.694–0.976], *p* = 0.025).

The ROC shows the diagnostic ability of a classifier as its discrimination threshold is varied. We performed a 10-fold cross validation and calculated the ROC to estimate the performance of our model (Figure 1). The performance of the hospitalization risk model was remarkably accurate, with an area under the curve (AUC) of 0.889. The model was able to predict 50%, 80%, and 90% of hospitalizations, with respective specificities of 95.3%, 82.2%, and 70.2%.

### 3.2. Factors Associated with Mortality for SARS-CoV-2-Positive Individuals

Table 3 displays the model for mortality risk. It is based on the comparison of 393 fatal cases of COVID-19 compared to 101,039 disease episodes that did not end in patient death. The smaller size of the outcome group limits the power of the model; nevertheless, a few factors are associated with a large statistically significant effect, and advanced age is even more strikingly associated with increased mortality risk. Each year of age increased the odds for death by a factor of 1.105. Compared to an individual aged 20 with similar other risk factors, a patient aged 60 is 54 times more likely to die of the disease, and a patient aged 80 is 393 times more likely to die. Being female is also associated with a reduced risk of death, reducing this risk by about half.

Being extremely obese and underweight both increased the risk of death (OR = 1.963 for a BMI of above 35 and OR = 2.179 for a BMI of below 18.5, compared to a normal BMI), while other BMI categories were not significantly associated with death. Impaired kidney function was also associated with an augmented mortality risk in a gradual manner (OR = 2.000 for a GFR of between 45 and 59, OR = 3.097 for a GFR of between 30 and 44, and OR = 6.888 for a GFR of below 30 compared to the reference category of a GFR of above 90, *p* < 0.001 for all). Diabetes mellitus, as reflected by the last hemoglobin A1C, also increased mortality risk in a gradual manner, although more moderately than for hospitalization. The other comorbidity coefficients are, overall, similar to those for hospitalization, although the smaller outcome group allowed us to detect a statistical significance only for hypertension and pulmonary disease. Vaccination with a booster dose significantly decreased mortality risk by 78% (OR = 0.223, 95% CI [0.091–0.551], *p* = 0.001).

We performed a 10-fold cross validation and plotted the ROC to estimate the performance of the mortality risk model (Figure 2). The model was very accurate, with an AUC of 0.967, and was able to predict 50%, 80%, and 90% of death events, with respective specificities of 98.6%, 95.2%, and 91.2%.

### 3.3. Risk Calculators

Table 2 and Table 3 provide all the coefficients, as well as the formula that is used to calculate the absolute risk of a given individual, using the regression models described above. Basically, the coefficients are to be multiplied by the corresponding variables and summed to obtain the natural logarithm of the odds ratio. After exponentiation, the odds ratio can be converted to a probability by dividing it by its value plus one. This calculator is available online and can be used to calculate hospitalization and mortality probabilities for any given individual.

## 4. Discussion

This study developed models that can estimate the risks of subsequent hospitalization and death for any individual newly infected with SARS-CoV-2 using health records from a large healthcare provider in Israel. On 30 December 2021, the first batch of Pfizer’s Paxlovid anti-COVID-19 medication arrived in Israel. We were immediately faced with the practical question: which COVID-19 patients should be prioritized to receive Paxlovid? These models are being used to answer this question rapidly and fairly by providing estimates of the hospitalization and mortality risks for each newly infected patient using information extracted from electronic medical records—notably, the patient’s age, sex, number of vaccine doses received so far; baseline BMI, Hemoglobin A1C, and estimated glomerular filtration ranges; and the presence of hypertension, immune deficiency, and/or pulmonary disease diagnoses. These calculated risk estimates are remarkably accurate and help identify which patients are at high risk of severe and potentially lethal disease and should, therefore, be prioritized for early antiviral treatment.

Our study capitalizes on the centrally managed, comprehensive electronic health records maintained in our health organization that are continuously updated with hospitalization and death from COVID-19; the early adoption of a homogenous vaccination program in Israel, and of the booster dose; and the relatively large number of individuals that have tested positive for COVID-19. We deliberately opted for model simplicity by limiting the number of input variables to recognized clinical factors associated with disease severity that are documented in most health organizations. We intentionally did not include country-specific demographic variables to generate a model that is generalizable to other populations. Even with these limited inputs, the resulting AUCs are highly accurate, achieving 0.889 for hospitalization and 0.967 for mortality. The better AUC performance of the model for mortality risk likely reflects that death is mostly determined by the patient’s health status, while the decision to hospitalize a patient is additionally impacted by the availability of family or social supports at home and the patient’s own preferences, which are not accounted for in our model. Importantly, these mortality and hospitalization risk estimates can be given as soon as the disease is diagnosed, allowing us to identify which patients are most at risk for a severe outcome so that they can be given treatment early in the infection. The models can also prioritize which populations to vaccinate or to urge to receive booster doses to maximize lives saved and reduce the load on hospitalization facilities. Several attempts have already been made to build predictive models for COVID-19 severity, notably [4,9,19,35,36,37,38]. Among the five models cited in the preceding sentence, only the models of Iannou and colleagues [4] and of Experton and colleagues [27] predict at least one of risk of hospitalization or risk of death in a newly infected individual. However, these two models achieve much lower AUCs than our models and do not take vaccination into account. Furthermore, both models use more than 10 variables, some of which are non-clinical variables that are specific to the United States. Thus, there is still an unmet need for simple and internationally applicable models, which are analogous to CURB-65 [39], and could allow the instant triage of new patients using few essential parameters, including vaccination status. This approach allowed us to produce models with remarkably high accuracy.

Our study also provides further large-scale confirmation of recently published studies that showed that a third booster vaccine provides a sharp and almost immediate increase in protection [35,36]. The risk reduction from each additional vaccine dose rigorously and quantitively substantiates the public health message that the primary benefits of current COVID-19 vaccinations are protecting against death and severe disease, and protection against any infection is a secondary goal [40]. Since vaccination has also been shown to substantially decrease the risk for symptomatic infection, its overall cumulative effect on hospitalization and death are even greater than the odds ratio reported here.

Importantly, in contrast to what we and others found for the infection risk [30], we did not observe that time elapsed since vaccination significantly increased hospitalization and mortality risks. For these outcomes, the protective effect of vaccination was largely determined by the cumulative number of vaccine doses received. This may indicate that the immune system response elicited by mRNA vaccine injection has a more lasting effect on hospitalization and mortality risk than on the risk of symptomatic infection following exposure to the virus.

Our study has several limitations. First, it is based on a population which was vaccinated almost exclusively with the Pfizer/Biotech BNT162b2 vaccine, with the first two doses spaced by 21 days. It is uncertain how the estimated effect of vaccination under these conditions would apply to populations vaccinated using different vaccines or using a different vaccination schedule. Moreover, factors specific to our health organization may have affected the results, such as the level of education, ease of access to care, ethnicity, criteria for hospital admission, and treatment decisions that influence mortality. Evolving patient management policies could have a confounding effect on the number of vaccine doses at different times. In addition, the data on which our models rely were collected mostly before the Omicron variant of SARS-CoV-2 emerged. Initial reports suggest that infections with the Omicron variant may be less severe [41,42], and so hospitalization and death risks for Omicron may be lower than those calculated by our models. Nevertheless, as long as Omicron is affected by the same risk factors as previous variants, the ranking of patients by risk is expected to remain similar. In our health organization, we use these models as tools to identify new COVID-19 patients who are most at risk for severe disease and could therefore benefit from the new antiviral treatments. We continuously monitor the hospitalization and mortality outcomes of COVID-19 infected individuals, and if we identify a different effect of risk factors specific to new viral variants, we will update our models accordingly. Additional studies on different populations would help to ascertain the validity of our models in different settings. To enable such validation studies, we provide the full model formulas and encourage their use.

In conclusion, the models described here, available online as a calculator, allow for the identification of individuals most at risk for severe disease or death if infected by using very few essential parameters and vaccination status. This approach can guide public health decisions to optimally allocate vaccines and scarce medicines to maximize lives saved [5].

## Figures and Tables

**Figure 1 microorganisms-10-01238-f001:**
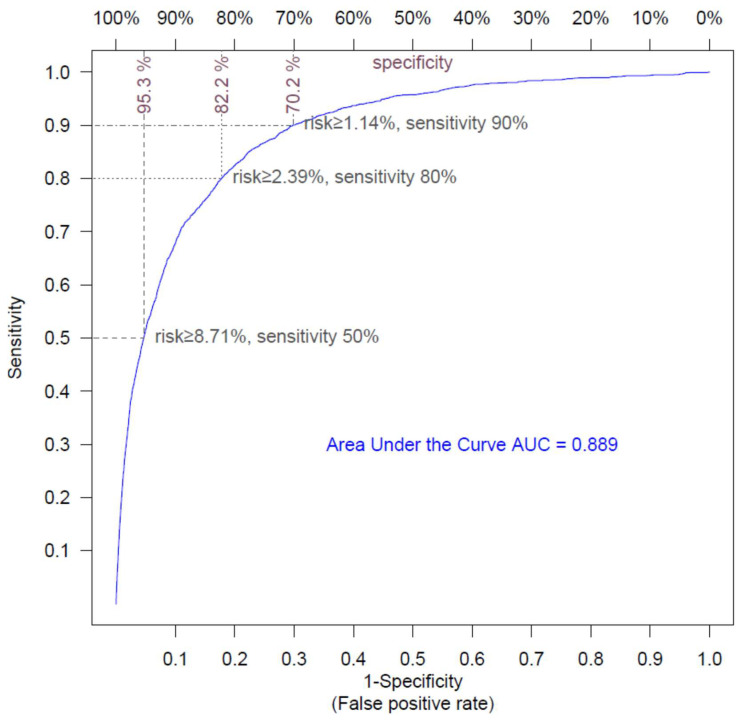
Receiver operating curve for the hospitalization risk model. The ROC shows the sensitivity and the specificity of the hospitalization model as its discrimination threshold is varied. With a threshold of 8.71% for risk, 50% of the COVID-19 episodes necessitating hospitalization can be identified (sensitivity = 50%), and specificity is 95.3% (false positive rate = 4.7%); with a risk threshold of 2.39%, sensitivity is 80% and specificity is 82.2% (false positive rate = 7.8%); and with a risk threshold of 1.14%, sensitivity is 90% and specificity is 70.2% (false positive rate = 30.8%).

**Figure 2 microorganisms-10-01238-f002:**
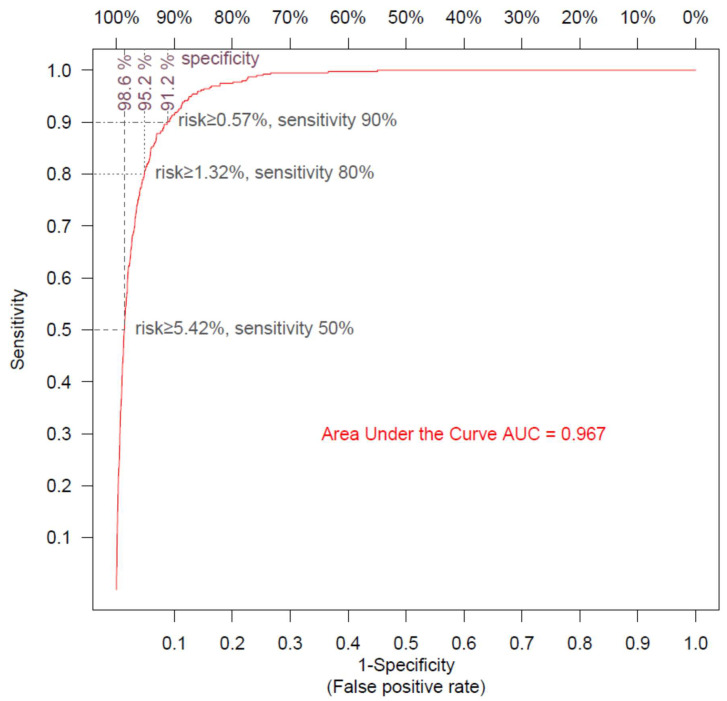
**Receiver operating curve for mortality risk model.** The ROC shows the sensitivity and the specificity of the mortality model as its discrimination threshold is varied. With a threshold of 5.42% for risk, 50% of the COVID-19 episodes ending in patient death can be identified (sensitivity = 50%), and specificity is 98.6% (false positive rate = 1.4%); with a risk threshold of 1.32%, sensitivity is 80% and specificity is 95.2% (false positive rate = 4.8%); and with a risk threshold of 0.57%, sensitivity is 90% and specificity is 91.2% (false positive rate = 8.8%).

**Table 1 microorganisms-10-01238-t001:** (**A**) Demographic and clinical characteristics of the study population. (**B**) Clinical characteristics of the study population after missing variables imputation.

**(A)**					
			**Not Hospitalized**	**Hospitalized** **(Not Deceased)**	**Deceased**
**N (%)**			98,894 (97.9%)	1752 (1.7%)	393 (0.4%)
**Vaccines doses**	**0**		82,261 (83.2%)	1405 (80.2%)	295 (75.1%)
	**1**		4732 (4.8%)	138 (7.9%)	32 (8.1%)
	**2**		10,436 (10.6%)	176 (10.0%)	61 (15.5%)
	**3**		1465 (1.5%)	33 (1.9%)	5 (1.3%)
**Sex**	**Female**		48,565 (49.1%)	798 (45.5%)	169 (43.0%)
**Age**	**Mean (SD)**		29.44 (19.17)	58.44 (19.03)	75.27 (13.06)
**Age category**	**5–11**		19,603 (19.8%)	19 (1.1%)	0 (0.0%)
	**12–17**		15,999 (16.2%)	30 (1.7%)	0 (0.0%)
	**18–39**		34,374 (34.8%)	245 (14.0%)	6 (1.5%)
	**40–59**		20,361 (20.6%)	561 (32.0%)	44 (11.2%)
	**≥60**		8557 (8.7%)	897 (51.2%)	343 (87.3%)
**Comorbidities**	**Hypertension**		9321 (9.4%)	880 (50.2%)	301 (76.6%)
	**Pulmonary disease**		1592 (1.6%)	167 (9.5%)	74 (18.8%)
	**Malignancy**		2258 (2.3%)	202 (11.5%)	83 (21.1%)
**BMI category**	**Underweight**	**<18.5**	22,506 (22.8%)	44 (2.5%)	10 (2.5%)
	**Normal**	**18.5–25**	32,373 (32.7%)	283 (16.2%)	83 (21.1%)
	**Overweight**	**25–30**	21,396 (21.6%)	566 (32.3%)	126 (32.1%)
	**Obese I**	**30–35**	10,763 (10.9%)	444 (25.3%)	85 (21.6%)
	**Obese II+**	**≥35**	5493 (5.6%)	372 (21.2%)	72 (18.3%)
	***missing***		6363 (6.4%)	43 (2.5%)	17 (4.3%)
**Kidney function GFR category**	**G1 (normal)**	**≥90**	64,097 (64.8%)	850 (48.5%)	95 (24.2%)
	**G2**	**60–89**	12,622 (12.8%)	596 (34.0%)	142 (36.1%)
	**G3a**	**45–59**	840 (0.8%)	140 (8.0%)	68 (17.3%)
	**G3b**	**30–44**	263 (0.3%)	74 (4.2%)	46 (11.7%)
	**G4/G5**	**<30**	151 (0.2%)	45 (2.6%)	37 (9.4%)
	***missing***		20,921 (21.2%)	47 (2.7%)	5 (1.3%)
**Hemoglobin A1C** **range**	**<6.5**		38,743 (92.2%)	1106 (72.2%)	268 (70.7%)
	**[6.5, 8.0]**		2129 (5.1%)	253 (16.5%)	70 (18.5%)
	**[8.0, 10.0]**		815 (1.9%)	115 (7.5%)	31 (8.2%)
	**≥10.0**		328 (0.8%)	58 (3.8%)	10 (2.6%)
	***missing***		56,879 (57.5%)	220 (12.6%)	14 (3.6%)
**(B)**					
			**Not Hospitalized**	**Hospitalized (Not Deceased)**	**Deceased**
**BMI category**	**Underweight**	**<18.5**	25,090 (25.4%)	47 (2.7%)	10 (2.5%)
	**Normal**	**18.5–25**	34,615 (35.0%)	294 (16.8%)	85 (21.6%)
	**Overweight**	**25–30**	22,687 (22.9%)	593 (33.8%)	139 (35.4%)
	**Obese I**	**30–35**	11,006 (11.1%)	446 (25.5%)	87 (22.1%)
	**Obese II+**	**≥35**	5496 (5.6%)	372 (21.2%)	72 (18.3%)
**Kidney function GFR category**	**G1 (normal)**	**≥90**	84,503 (85.4%)	887 (50.6%)	90 (22.9%)
	**G2**	**60–89**	13,124 (13.3%)	590 (33.7%)	144 (36.6%)
	**G3a**	**45–59**	860 (0.9%)	141 (8.0%)	71 (18.1%)
	**G3b**	**30–44**	256 (0.3%)	87 (5.0%)	47 (12.0%)
	**G4/G5**	**<30**	151 (0.2%)	47 (2.7%)	41 (10.4%)
**Hemoglobin A1C** **range**	**<6.5**		95,481 (96.5%)	1322 (75.5%)	282 (71.8%)
	**[6.5, 8.0]**		2266 (2.3%)	257 (14.7%)	70 (17.8%)
	**[8.0, 10.0]**		819 (0.8%)	115 (6.6%)	31 (7.9%)
	**≥10.0**		328 (0.3%)	58 (3.3%)	10 (2.5%)

**Table 2 microorganisms-10-01238-t002:** Logistic regression model for hospitalization risk.

			Odds Ratio *	95% Confidence Interval	*p*	β_i_ (Coefficient)
**β_0_**	**(Intercept)**		0.001		0.0000	−6.754369
**Age**	**Continuous in years**		1.061	[1.057–1.064]	0.0000	0.058834
**Sex**	**Male**		1.000	reference		0
	**Female**		0.657	[0.598–0.722]	0.0000	−0.420262
**Vaccine doses**	**0**		1.000	reference		0
	**1**		0.823	[0.694–0.976]	0.0248	−0.195301
	**2**		0.602	[0.521–0.697]	0.0000	−0.506982
	**3**		0.339	[0.241–0.476]	0.0000	−1.082553
**BMI category**	**Underweight**	**<18.5**	0.937	[0.697–1.260]	0.6674	−0.064998
	**Normal**	**18.5–25**	1.000	reference		0
	**Overweight**	**25–30**	1.324	[1.158–1.513]	0.0000	0.280302
	**Obese I**	**30–35**	1.664	[1.441–1.922]	0.0000	0.509396
	**Obese II+**	**≥35**	2.932	[2.514–3.419]	0.0000	1.075528
**Kidney function** **GFR category**	**G1 (Normal)**	**≥90**	1.000	reference		0
	**G2**	**60–89**	1.058	[0.947–1.183]	0.3197	0.056446
	**G3a**	**45–59**	1.568	[1.296–1.898]	0.0000	0.450065
	**G3b**	**30–44**	2.774	[2.164–3.555]	0.0000	1.020266
	**G4/G5**	**<30**	4.000	[2.952–5.420]	0.0000	1.386290
**Hemoglobin A1C** **%**	**<6.5**		1.000	reference		0
	**[6.5, 8.0]**		1.454	[1.263–1.673]	0.0000	0.374131
	**[8.0, 10.0]**		1.908	[1.559–2.334]	0.0000	0.645939
	**≥10.0**		3.048	[2.284–4.068]	0.0000	1.114620
**Comorbidities**	**Hypertension**		1.270	[1.130–1.428]	0.0001	0.239212
	**Pulmonary disease**		1.331	[1.134–1.563]	0.0005	0.286110
	**Malignancy**		1.197	[1.030–1.390]	0.0188	0.179418

* Odds ratio is defined as exp (coefficient). The coefficients in the last column are the β_i_ to be used to calculate the odds ratio using the following formula: odds ratio = exp (β_0_ + x_1_ β_1_ + x_2_ β_2_ + x_3_ β_3_ + x_4_ β_4_ + …). The probability of an event can be obtained from the odds ratio using the formula: *p* = (odds ratio)/(1 + odds ratio).

**Table 3 microorganisms-10-01238-t003:** Logistic regression model for mortality risk.

			Odds Ratio *	95% Confidence Interval	*p*	β_i_ (Coefficient)
**β_0_**	**(Intercept)**		0.000		0.0000	–11.227376
**Age**	**Continuous in years**		1.105	[1.095–1.115]	0.0000	0.099573
**Sex**	**Male**		1.000	reference		0
	**Female**		0.500	[0.401–0.625]	0.0000	–0.692446
**Vaccine doses**	**0**		1.000	reference		0
	**1**		0.921	[0.627–1.354]	0.6771	–0.081842
	**2**		0.936	[0.698–1.254]	0.6561	–0.066541
	**3**		0.223	[0.091–0.551]	0.0011	–1.498783
**BMI category**	**Underweight**	**<18.5**	2.179	[1.056–4.496]	0.0350	0.778997
	**Normal**	**18.5–25**	1.000	reference		0
	**Overweight**	**25–30**	0.979	[0.733–1.307]	0.8866	−0.021027
	**Obese I**	**30–35**	1.085	[0.785–1.500]	0.6196	0.081961
	**Obese II+**	**≥35**	1.963	[1.383–2.786]	0.0002	0.674479
**Kidney function** **GFR category**	**G1 (Normal)**	**≥90**	1.000	reference		0
	**G2**	**60–89**	1.283	[0.965–1.705]	0.0861	0.249162
	**G3a**	**45–59**	2.000	[1.390–2.878]	0.0002	0.693180
	**G3b**	**30–44**	3.097	[2.035–4.715]	0.0000	1.130578
	**G4/G5**	**<30**	6.888	[4.389–10.810]	0.0000	1.929831
**Hemoglobin A1C** **%**	**<6.5**		1.000	reference		0
	**[6.5, 8.0]**		1.137	[0.851–1.518]	0.3842	0.128408
	**[8.0, 10.0]**		1.479	[0.983–2.226]	0.0602	0.391618
	**≥10.0**		1.782	[0.905–3.510]	0.0948	0.577767
**Comorbidities**	**Hypertension**		1.348	[1.011–1.797]	0.0421	0.298497
	**Pulmonary disease**		1.475	[1.113–1.956]	0.0069	0.388824
	**Malignancy**		1.138	[0.868–1.491]	0.3489	0.129199

* Odds ratio is defined as exp (coefficient). The coefficients in the last column are the β_i_ to be used to calculate the odds ratio, using the following formula: odds ratio = exp (β_0_ + x_1_ β_1_ + x_2_ β_2_ + x_3_ β_3_ + x_4_ β_4_ + …). The probability of an event can be obtained from the odds ratio using the formula: *p* = (odds ratio)/(1 + odds ratio).

## Data Availability

This study is based on real-world patient data including comorbidity factors that cannot be communicated due to patient privacy concerns.

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
