# Peer review of "A Calculator for COVID-19 Severity Prediction Based on Patient Risk Factors and Number of Vaccines Received"

_microorganisms, 2022, doi:10.3390/microorganisms10061238_

Round 1

Reviewer 1 Report

In this paper, the authors describe a mdel enable to identify patients at high risk of hospitalization and death when infected by SARS-CoV-2. My major comment is that the model described is based on data from Leumit Health Services (LHS). How are these data representative of the general population, is there not a risk of bias (level of education, ease of access to care, etc.). A description of the patients who join this group is essential, and this should figure as a potential limitation in the discussion.

Author Response

We are pleased that reviewer 1 appreciates the importance of this study and its methodology.

We thank the reviewer for his suggestion to describe the Leumit Health Services  (LHS) population on which this study is based and add this as a potential limitation of the study.

Table 1 is a descriptive table showing detailed demographic and clinical characteristics of the LHS population on which this study is based. Moreover, in the beginning of the Methods section, we provide a short description of the LHS organization, to which we have added the fact that "All LHS members have similar comprehensive health insurance and similar access to healthcare services, as determined by Israel's ministry of health for the national healthcare providers. " (Line 69).

In addition, in the discussion, we have made the limitations more explicit on this issue: "Moreover, factors specific to our health organization may have affected the results, such as the level of education, ease of access to care, ethnicity, criteria for hospital admission, and treatment decisions that influence mortality. " (Line 292).

Reviewer 2 Report

This manuscript covers an interesting topic. 

It offers an important analysis of factors associated with a higher risk of hospitalization and death because of COVID. Moreover, it provides the possibility to estimate an individual's risk of hospitalization and death contributing to prioritizing populations that need booster doses or vaccine priority. 

The introduction is concise but offers a good overview of the background.

The methods are well described and the results are well organized and reported.

Proving the link for the calculator is of paramount importance, please consider making it more clear in the abstract and in the title.

The discussion is well structured. Immediately after line 235, please consider adding a very short summary of the main results (one/two brief sentences).

When referring to previous similar studies, please consider also refer to doi: 10.3389/ijph.2022.1604427; doi: 10.1016/j.medine.2021.04.014

Author Response

We are pleased that reviewer 2 appreciates the topic, the importance of this study, and the methodology. We are also grateful for his suggestions, that we have followed:

Proving the link for the calculator is of paramount importance, please consider making it more clear in the abstract and in the title.

Response: Thanks. We have now changed the title to: A Calculator for COVID-19 Severity Prediction Based on Patient Risk Factors and Number of Vaccines Received.

---

The discussion is well structured. Immediately after line 235, please consider adding a very short summary of the main results (one/two brief sentences).

Response: Thanks for this suggestion. We have followed this suggestion and added the following text in the discussion (beginning at  line 236):

These models are being used to answer this question rapidly and fairly, by providing estimates of the hospitalization and mortality risks for each newly infected patient, using information extracted from electronic medical records, notably the patient's age, sex, number of vaccine doses received so far; baseline BMI, HBA1C, and estimated glomerular filtration ranges; and the presence of hypertension, immune deficiency, and/or pulmonary disease diagnoses. These calculated risk estimates are remarkably accurate and help identify which patients are at high risk of severe and potentially lethal disease and should therefore be prioritized for early antiviral treatment.

When referring to previous similar studies, please consider also refer to

doi: 10.3389/ijph.2022.1604427;

doi: 10.1016/j.medine.2021.04.014

Response: Thanks. We have added references to the two suggested studies (References 37 and 38)